# Contemporary Molecular Markers for Predicting Systemic Treatment Response in Urothelial Bladder Cancer: A Narrative Review

**DOI:** 10.3390/cancers16173056

**Published:** 2024-09-01

**Authors:** George Dimitrov, Radoslav Mangaldzhiev, Chavdar Slavov, Elenko Popov

**Affiliations:** 1Department of Medical Oncology, Medical University of Sofia, University Hospital “Tsaritsa Yoanna”, 1527 Sofia, Bulgaria; gdimitrov@medfac.mu-sofia.bg (G.D.); rmangaldziev@medfac.mu-sofia.bg (R.M.); 2Department of Urology, Medical University of Sofia, University Hospital “Tsaritsa Yoanna”, 1527 Sofia, Bulgaria; ch.k.slavov@gmail.com

**Keywords:** urothelial carcinoma, biomarkers, precision oncology, response predictors

## Abstract

**Simple Summary:**

Bladder cancer is a serious and challenging disease to treat. Researchers are actively seeking new ways to predict how well different treatments will work for each patient. This review examines the latest research on specific tumor markers, known as molecular biomarkers, that can help doctors anticipate how a patient’s cancer will respond to treatment. By focusing on these biomarkers, the goal is to enhance personalized medicine, where treatments are tailored to each patient’s unique cancer profile. This approach could lead to better outcomes, allowing patients to receive the most effective treatments with fewer side effects.

**Abstract:**

The search for dependable molecular biomarkers to enhance routine clinical practice is a compelling challenge across all oncology fields. Urothelial bladder carcinoma, known for its significant heterogeneity, presents difficulties in predicting responses to systemic therapies and outcomes post-radical cystectomy. Recent advancements in molecular cancer biology offer promising avenues to understand the disease’s biology and identify emerging predictive biomarkers. Stratifying patients based on their recurrence risk post-curative treatment or predicting the efficacy of conventional and targeted therapies could catalyze personalized treatment selection and disease surveillance. Despite progress, reliable molecular biomarkers to forecast responses to systemic agents, in neoadjuvant, adjuvant, or palliative treatment settings, are still lacking, underscoring an urgent unmet need. This review aims to delve into the utilization of current and emerging molecular signatures across various stages of urothelial bladder carcinoma to predict responses to systemic therapy.

## 1. Introduction

Globally, urothelial bladder cancer (UBC) accounts for approximately 570,000 new cases and over 200,000 deaths annually [1]. Despite advancements in detection, treatment, and surveillance methods, the overall prognosis for UBC remains largely unchanged [2]. Efforts to improve outcomes through the integration of novel therapeutics and enhanced surveillance protocols have proven ineffective in reducing mortality rates associated with UBC [3]. This emphasizes the critical need for further research and innovative approaches to effectively manage this disease.

Early urothelial bladder cancer manifests in two distinct phenotypes: non-muscle-invasive (NMIBC), where cancer cells are confined within the urothelial layer, and muscle-invasive (MIBC), characterized by infiltration beyond the subepithelial connective tissue [4]. The progression of bladder cancer through these stages involves genetic alterations that impact crucial cellular survival pathways, ultimately leading to advanced systemic disease [5]. Notably, studies have revealed that poorly-differentiated NMIBC and MIBC share similar genetic signatures [6,7]. This suggests that despite their differing initial clinical behavior, there are underlying genetic commonalities between these currently considered separate phenotypes of UBC. Understanding these genetic signatures could offer valuable insights into future treatment strategies.

Radical cystectomy (RC) with pelvic lymph node dissection remains a cornerstone treatment for MIBC. However, in the past decade, the only systemic treatment that has demonstrated additional curative potential benefit is cisplatin-based therapy, offering a modest absolute survival benefit of around 10% when applied in the neoadjuvant setting [8]. However, despite evidence supporting its efficacy [9,10], neoadjuvant chemotherapy (NACT) remains underutilized, likely due to factors such as advanced age at diagnosis, existing comorbidities, and treatment toxicity [11]. Therefore, additional stratification factors are needed to further optimize patient outcomes.

Contemporary personalized and precision medicine approaches rely on clinically informative and easily obtainable molecular signatures. However, UBC lags behind other solid malignancies, such as lung cancer, in the integration of biomarkers into guidelines and clinical decision-making processes [12]. Current research in UBC predominantly focuses on two biomarker avenues: tissue-based (tumor genetic and protein expression profiles) and biological fluid-based (urine and serum biomarkers) [13].

Tissue-based biomarkers enable comprehensive molecular tumor profiling, aiding in the identification of patients at risk of recurrence and predicting tumor responses to various therapeutic modalities. This profiling includes assessing the status of DNA damage response and repair (DDR) genes, identifying driver mutations such as FGFR3, VEGF-C, GATA3, FOXA1, and TP53, and evaluating the predictive value of PD-L1/PD-1, among other factors [14].

Urine biomarker research primarily targets early-stage disease and focuses on detecting and analyzing exfoliated bladder cancer cells (EBCCs), exosomes, and cell-free DNA (cfDNA) to facilitate a non-invasive method for initial disease diagnosis and monitoring [15].

Serum-based markers detect circulating cancer cells, genetic material, or specific genes. The discourse surrounding these blood-based liquid biopsies primarily revolves around circulating tumor cells (CTCs), extracellular vesicles or exosomes, and cell-free nucleic acids: cfDNA, cell-free RNA (cfRNA), and cell-free microRNA (cfmiRNAs). These blood-based liquid biopsies may prove particularly beneficial for post-cystectomy patients, where urinary biomarkers are less applicable [16,17].

This narrative review explores the evolving roles of molecular biomarkers in urothelial bladder carcinoma. With the increasing need for biomarker-guided therapies, including immune checkpoint inhibitors (ICIs) and targeted treatments in advanced and metastatic stages, understanding the landscape of molecular biomarkers is crucial.

## 2. Methodology

This review was conducted using comprehensive searches in PubMed, Scopus, and Web of Science databases. The primary keywords employed included ‘urothelial carcinoma’, ‘bladder cancer’, ‘urothelial carcinoma’, ‘molecular markers’, ‘treatment response’, ‘prognostic biomarkers’, ‘predictive biomarkers’, ‘systemic therapy’, ‘immune checkpoint inhibitors’, and ‘targeted therapies’. These keywords were combined using Boolean operators (AND, OR) to ensure a thorough search that encompassed all relevant studies.

We included articles published between January 2000 and January 2024, focusing on original research, clinical trials, and meta-analyses that provided robust data on molecular markers in UBC (Figure 1).

Inclusion criteria were as follows:Studies that specifically addressed molecular biomarkers in UBC.Articles discussing biomarkers related to systemic treatment response or prediction of therapeutic outcomes.Research that provided significant clinical or experimental data supporting the role of these biomarkers.Publications in peer-reviewed journals, available in English.Exclusion criteria were as follows:Studies focused exclusively on non-muscle-invasive bladder cancer (NMIBC).Articles lacking sufficient experimental or clinical data to support their findings.Reviews, editorials, duplicates, and case reports without new data or substantial contribution to the field.

After selecting articles meeting the set criteria, we extracted and categorized data on molecular biomarkers, treatment responses, and prognostic factors. We performed a qualitative synthesis to identify common themes and trends, complemented by quantitative analysis for meta-analytic data based on the robustness of the data and the validity of experimental or clinical methodologies. This approach allowed us to integrate findings, assess consistency across studies, and highlight key biomarkers with clinical and statistical relevance.

## 3. Pathology of Bladder Tumors

Traditionally, urothelial bladder cancer (UBC) has been categorized into two primary groups: non-muscle-invasive bladder cancer (NMIBC) and muscle-invasive bladder cancer (MIBC). NMIBC consists of tumors limited to the mucosa, categorized as non-invasive papillary carcinoma (pTa) and carcinoma in situ (CIS, pTis), or carcinomas invading the subepithelial connective tissue (pT1). This category constitutes a significant portion (~75%) of UBC cases and is characterized by frequent tumor recurrence, limited progression, and a higher survival rate with lower cancer-specific mortality [18].

On the other hand, MIBC comprises high-grade tumors that are either locally advanced, invading the muscularis propria (pT2), or extending into the perivesical soft tissue (pT3), as well as extravesical tumors involving adjacent organs or the pelvic and abdominal wall (pT4), or metastatic tumors (M1) [19]. Current recommendations for the treatment of MIBC involve cisplatin-based NACT followed by radical cystectomy or trimodal therapy (i.e., maximal trans-urethral resection of the bladder tumor followed by concomitant chemoradiotherapy) in certain cases [12].

UBC is characterized by a diverse array of morphological and genomic variations, contributing to its classification as a highly heterogeneous disease. This heterogeneity is evident in the wide spectrum of subtype histologies observed in UBC patients, each with distinct characteristics and clinical behaviors. Additionally, at the molecular level, UBC exhibits considerable variability in genetic alterations, including mutations, copy number variations, and gene expression profiles [20].

## 4. Systemic Therapies in UBC

### 4.1. Platinum-Based Systemic Chemotherapy

Platinum derivatives are foundational in systemic therapy for UBC, acting as potent alkylating agents that impede tumor growth by disrupting essential DNA processes such as replication and transcription. Cisplatin, a key example, penetrates cells via membrane transporters, employing three primary mechanisms: alkylating DNA bases, creating cross-links to hinder DNA separation, and inducing nucleotide mispairing, potentially causing mutations. Platinum-based regimens are recommended in the neoadjuvant [9,21], adjuvant [22], and metastatic setting of UBC [23,24]. Despite its efficacy, resistance to cisplatin can manifest through various pathways, including pre-target, on-target, post-target, and off-target mechanisms, involving diverse genes, enzymes, and transporters [25]. Understanding these mechanisms not only illuminates the pharmacodynamics of cisplatin but also reveals potential biomarkers and therapeutic targets to enhance treatment outcomes.

### 4.2. Immune Checkpoint Inhibitors

Immune checkpoint inhibitors (ICIs) are monoclonal antibodies that target cell surface proteins such as programmed death-1 (PD1), programmed death-ligand 1 (PD-L1), and cytotoxic T-lymphocyte-associated antigen 4 (CTLA-4). By blocking these proteins, ICIs remove the inhibition of T cells, enhancing their cytotoxic activity against cancer cells. In the UBC setting, atezolizumab, an anti-PD-L1 agent, was the first ICI approved for advanced disease in 2016. Since then, other PD-L1 inhibitors (avelumab and durvalumab) and PD1 inhibitors (nivolumab and pembrolizumab), have demonstrated improved survival in pretreated patients with advanced UBC compared to chemotherapy [26]. Despite these advancements, the overall response rate and progression-free survival (PFS) for these agents remain limited [27]. Consequently, there is a pressing need to explore and develop new immune-enhancing treatment combinations and identify response predictors to improve survival outcomes for patients with UBC.

### 4.3. Target Therapies

Recent progress in comprehending the molecular characteristics of UBC has led to updated treatment approaches. FGFR, a receptor tyrosine kinase, represents a target in UBC, with erdafitinib, an oral pan-FGFR inhibitor, now approved for advanced UBC treatment [28]. Antibody–drug conjugates (ADCs) capitalize on highly expressed tumor proteins as drug delivery targets. Two ADCs have gained approval for treating UBC. Enfortumab vedotin targets Nectin-4, often overexpressed in UBC, and utilizes monomethyl auristatin E as the linked microtubule inhibitor. A global phase 3 trial involving 608 patients who had prior platinum-containing chemotherapy and PD-1/PD-L1 inhibitors demonstrated longer OS and PFS with enfortumab vedotin compared to chemotherapy [29]. The FDA subsequently approved enfortumab vedotin [30]. Sacituzumab govitecan, another ADC, targets Trop-2 with an anti-Trop-2 monoclonal antibody hRS7 IgG1κ combined with SN-38, the active metabolite of irinotecan [31]. The approvals of enfortumab vedotin and sacituzumab govitecan mark significant progress in advanced UBC treatment. Despite the introduction of these novel drugs, the overall response rate and clinical outcomes for these agents remain constrained.

### 4.4. Combining Systemic Modalities

Recently, a significant advancement occurred with the approval of pembrolizumab in combination with enfortumab vedotin as a first-line treatment for cisplatin-eligible patients [23]. In the phase 3 EV-302 trial, researchers compared the efficacy and safety of this combination to platinum-based chemotherapy in previously untreated locally advanced or metastatic urothelial carcinoma. Patients were randomly assigned to receive either enfortumab vedotin and pembrolizumab or chemotherapy, with primary endpoints being progression-free survival and overall survival. The results demonstrated that patients in the enfortumab vedotin–pembrolizumab group had significantly longer progression-free survival (median, 12.5 months vs. 6.3 months) and overall survival (median, 31.5 months vs. 16.1 months) compared to the chemotherapy group. Moreover, treatment-related adverse events of grade 3 or higher were less common in the enfortumab vedotin–pembrolizumab group. These findings establish enfortumab vedotin and pembrolizumab as superior to chemotherapy for untreated locally advanced or metastatic urothelial carcinoma, establishing this regimen as the recommended standard of care for advanced UBC in the first line [32].

### 4.5. Limitations of Current Therapies and Patient Selection

While advancements in systemic therapies for UBC offer promising treatment options, several limitations must be addressed. Patient selection is a critical factor influencing the maximal success of these therapies. Personalized approaches based on individual patient profiles, including genetic, molecular, and clinical characteristics, are necessary to enhance treatment efficacy and minimize adverse effects. The development and validation of predictive biomarkers are essential for guiding therapy choices and improving patient outcomes. Moreover, ongoing research is needed to refine patient selection criteria and optimize the use of these therapies in clinical trials and routine practice.

## 5. Tissue-Based Biomarkers

### 5.1. Molecular Subtypes of UBC

In addition to morphological diversity, UBC also displays genomic heterogeneity, with distinct molecular alterations observed across different subtypes and individual tumors. Genetic aberrations, including mutations in oncogenes and tumor suppressor genes, chromosomal rearrangements, and alterations in DNA repair pathways, contribute to the development and progression of UBC. Furthermore, advancements in genomic profiling techniques have revealed intricate molecular subtypes of UBC, each associated with specific biological characteristics and clinical implications [33]. Recent studies have explored the use of surrogate markers for molecular classification of bladder carcinoma using paraffin-embedded tissue, which is commonly available in clinical settings. These surrogate markers can facilitate the molecular subtyping of UBC without the need for fresh frozen tissue, making it easier to integrate molecular classification into routine practice [34,35].

The molecular heterogeneity of UBC encompasses six subtypes: luminal papillary, luminal nonspecified, luminal unstable, stroma-rich, basal/squamous, and neuroendocrine-like [36]. Each subtype presents unique features in terms of cellular morphology, tissue architecture, and differentiation patterns (Table 1). These variations contribute to differences in disease progression, treatment response, and patient outcomes [7].

The basal subtype, characterized by high KRT5/6 and KRT14 and low FOXA1 and GATA3 expression, responds well to NACT [37]. In contrast, the p53-like subtype shows poor response and worse survival rates while luminal papillary tumors have the best prognosis regardless of treatment [38]. These findings were validated in a phase II trial of NACT with dose-dense MVAC and bevacizumab, highlighting the chemoresistance of p53-like tumors [39].

Whole transcriptome profiling of MIBC patients treated with various NACT regimens revealed that basal subtypes benefit from NACT, showing improved OS. Claudin-low subtypes, marked by immune infiltration and immunosuppression, had poor OS irrespective of treatment [40]. These tumors had increased mutations in RB1, EP300, and NCOR1 and decreased mutations in FGFR3, ELF3, and KDM6A, suggesting potential responsiveness to ICIs [41].

Neuroendocrine (NE)-like tumors, characterized by neuronal-associated gene expression, had worse disease-specific survival (DSS) but similar pathological downstaging rates compared to other subtypes [42].

An analysis of tissue microarrays revealed that NACT induces an epithelial–mesenchymal transition (EMT) phenotype, with increased mesenchymal markers predicting poor outcomes [43].

Understanding these molecular subtypes and their responses to therapy highlights the importance of personalized treatment strategies in UBC, reflecting the tumor’s adaptability and the complex interplay between disease, the immune system, and therapeutic stress.

### 5.2. DNA Damage Response and Repair (DDR) Genes

DDR genes play a crucial role in repairing DNA damage caused by platinum-based agents, such as cisplatin, via pathways like nucleotide excision repair (NER) [44]. Deficiencies in these genes, often due to non-synonymous mutations, can increase sensitivity to platinum-based therapy. Among these genes, ERCC1 and ERCC2 are significant regulators of the NER process. While studies on ERCC1 have yielded conflicting results [45], ERCC2 has been extensively researched [46]. Mutations in ERCC2 have been associated with favorable responses to platinum-based neoadjuvant chemotherapy and improved overall survival (OS) in metastatic bladder cancer patients and with improved pathologic downstaging (pDS) in the neoadjuvant setting [21,47]. Additionally, a three-gene signature involving ATM, RB1, and FANCC has shown promise in predicting response to NACT and improving progression-free survival (PFS), disease-specific survival (DSS), and OS (Table 2) [48]. Other DDR genes, such as FANCD2, PALB2, BRCA1, and BRCA2, have also been linked to recurrence-free survival (RFS) in MIBC patients [49]. Moreover, DDR alterations have shown a correlation with response to ICIs in metastatic UBC patients [50]. In the recently published results from the RETAIN trial, which aimed to tailor MIBC treatment based on DDR gene alterations, the 2-year metastasis-free survival (MFS) rate of 72% did not meet the pre-specified non-inferiority criterion [51]. While DDR alterations hold promise, further research and validation are needed to integrate them into routine clinical management effectively.

### 5.3. Driver Mutations

Driver mutations in FGFR3, ERBB1/2, and PIK3Ca have emerged as significant predictors in patients with MIBC undergoing various NACT regimens (Table 3) [52,53].

Alterations in FGFR3 are found in around 15% of urothelial carcinomas, with a higher frequency in upper-tract urothelial carcinoma (UTUC) and luminal papillary tumors. These mutations, including S249C, R248C, and Y373C, result in continuous receptor dimerization and phosphorylation without the need for a ligand. This process activates several oncogenic pathways, especially the MAPK signaling pathway [54]. FGFR3 mutations are often mutually exclusive with TP53 and RB1 mutations but can co-occur with ERBB2 and KRAS alterations [52]. FGFR3 mutations have been associated with worse recurrence-free survival rates following perioperative cisplatin-based chemotherapy [55].

ErbB1/HER1, also known as EGFR, is a recognized oncogenic target in cancers such as non-small-cell lung cancer, glioblastoma, and basal-like breast cancers [56]. HER1 is positive in 75% of primary bladder cancers, with 86% concordance in metastatic lesions [57]. However, EGFR expression has not been established as an independent predictor of disease progression or poor overall survival [58].

HER2, or receptor tyrosine-protein kinase ERBB2, is part of the ErbB/HER family of receptor tyrosine kinases. HER2 overexpression activates multiple signaling pathways that promote proliferation and tumorigenesis [59]. While HER2 is a well-established target in breast and gastric cancers, its role in bladder cancer is less defined. Accurate HER2 assessment is crucial for effective therapy selection. HER2 overexpression occurs in 9.2–12.4% of invasive bladder carcinomas, with gene amplification in 5.1% of cases [60]. These alterations are more common in luminal subtypes and metastatic tumors than in basal subtypes and primary tumors [61].

The phosphatidylinositol 3-kinase (PI3K) pathway, involved in cell growth, tumorigenesis, cell invasion, and drug response, is frequently activated in urothelial bladder cancer (UBC) due to PIK3CA alterations. Mutations in PIK3CA are found in 20–30% of UBC cases and typically occur early in tumor development [62]. Predominantly, these mutations affect the helical domain (E545K/D or E542K), with fewer mutations in the catalytic domain (H1047R). There is no significant association between PIK3CA mutations and clinical characteristics, leaving the potential therapeutic benefit of PI3K pathway inhibition unclear [63].

Driver mutations in these genes can serve as predictive biomarkers for therapeutic response and are targetable with novel agents. For instance, erdafitinib, a pan-FGFR inhibitor, and afatinib, an oral irreversible inhibitor of the ERBB family, have shown efficacy in platinum-refractory metastatic patients with FGFR3 or ERBB alterations, respectively, successfully meeting primary endpoints in phase III clinical trials [28,64].

### 5.4. PD-L1/PD-1 Expression

The predictive value of PD-L1/PD-1 expression (TPS, CPS, TC, or IC) has been extensively studied and remains controversial as PD-L1/PD-1 biomarker evaluation suffers from heterogeneity in assays and cut-offs across studies (Table 4) [65]. Therefore, for the time being, in the UBC setting, no clinically applicable biomarker has succeeded in identifying patients that will benefit from PD-L1/PD-1 inhibition therapies consistently.

In the PURE-01 trial (NCT02736266), a phase II study investigating pembrolizumab as a neoadjuvant chemotherapy regimen, a pathologic complete response (pCR) rate of 42% was observed. Comprehensive genomic profiling and PD-L1 combined positive score (CPS, Dako 22C3 antibody, cut-off 10%) were assessed from pre- and post-therapy tissue samples. Later a linear association between increasing TMB and CPS values was demonstrated with pT0N0, indicating that higher TMB and CPS were significantly associated with complete response [66]. Based on these findings, a composite biomarker-based pT0N0 probability calculator was proposed.

In contrast, the ABACUS trial (NCT02662309), a phase II study testing atezolizumab before radical cystectomy in 95 cisplatin-ineligible patients with MIBC, showed differing results. This study indicated that pre-existing T-cell immunity, rather than TMB combined with DDR gene signatures, was the key factor in achieving pCR [67]. Both studies did agree on the importance of pre-existing immunity, as evidenced by CD8+ T-cell infiltration or immune-related gene signatures, in correlating with pathologic response.

Recently, five distinct genetic and transcriptomic programs were identified and validated in an independent neoadjuvant ICI trial to pinpoint features of response or resistance. Histone demethylase KDM5B is a repressor of tumor immune signaling pathways. Inhibiting KDM5B enhanced immunogenicity in FGFR3-mutated UBC cells [68]. Such findings suggest the possibility for additional molecular stratification for ICI response and therapeutic alternatives for resistant subtypes.

### 5.5. Tumor-Mutation Burden

Tumor mutational burden (TMB) refers to the total number of substitutions and insertions or deletions per one million bases in gene exons within tumor tissue [69]. Bladder cancer, with a high TMB (≥10 mutations per megabase), is particularly responsive to immunotherapy, notably ICIs [70]. Recently, EP300, a gene encoding an adenoviral E1A-binding protein that functions as a transcriptional co-factor and histone acetyltransferase (KAT), has been linked to higher tumor mutational burden. It is speculated that mutations in EP300 may boost immune response, playing a key role in the effectiveness of immune checkpoint inhibitor therapies [71].

Moreover, a retrospective study employing whole-exome sequencing (WES) and genomic variant call format (VCF) files assessed TMB to predict the response to cisplatin-gemcitabine neoadjuvant chemotherapy in patients with muscle-invasive bladder cancer (MIBC). The study found a significant correlation between high TMB and alterations in DNA damage response (DDR) genes, observed in 38.1% of cases. The most frequently mutated genes were TP53 (45%), ARID1A/B (40%), and KMT2B/C/D/E (35%). Although there was a trend toward higher TMB in patients who achieved pathologic downstaging (pDS), it was not statistically significant. Additionally, virtual karyotype analysis revealed that 71.4% of non-responders had an amplification of the chromosomal region 7p12. This region includes genes such as HUS1, ABCA13, EGFR, FIGNL1, and IKZF1, which are implicated in resistance to cisplatin [72].

### 5.6. Biomarker Interaction

The co-expression of mutations in MIBC significantly influences the mechanisms of aggressiveness and poor response to treatment. Co-alterations in TP53 and RB1 are associated with increased TMB, higher numbers of predicted neoantigens, and enhanced immune cell infiltration, including CD8+ T cells and NK cells. These tumors exhibit a higher response rate to immune checkpoint inhibitors like atezolizumab, suggesting that these co-mutations may enhance immunogenicity and responsiveness to PD-L1/PD-1 inhibition therapies [73].

Co-mutations in DDR genes, such as those involved in nucleotide excision repair (e.g., ERCC2), are linked to increased sensitivity to cisplatin-based chemotherapy. These mutations impair the tumor’s ability to repair DNA damage, thereby enhancing the efficacy of DNA-damaging agents [74]. Additionally, DDR gene mutations are associated with altered expression of immune regulatory genes, which may contribute to immune evasion and impact the response to immunotherapies [75].

Mutations and overexpression in the PPARγ/RXRα pathway can lead to immune evasion by inhibiting CD8+ T-cell infiltration and cytokine expression. This pathway’s activation is associated with resistance to immunotherapies, highlighting its role in tumor aggressiveness and poor treatment response [76].

### 5.7. miRNAs

MicroRNAs (miRNAs) have gained attention as potential biomarkers for diagnosing and predicting survival in patients with urothelial bladder cancer. These small, noncoding RNA molecules regulate gene expression by binding to the 3′-untranslated region (3′-UTR) of mRNA, thereby inhibiting its function. They can be easily isolated from patient samples like urine and serum and evaluated using real-time quantitative reverse-transcription PCR (RT-qPCR). Several miRNAs have been implicated in UBC resistance and recurrence, offering potential insights into patient management (Table 5) [77,78]. Despite promising findings, the definitive role of miRNAs in predicting treatment response in UBC is still under investigation. However, these results provide a basis for exploring new therapeutic targets.

## 6. Biological Fluid-Based Biomarkers

Liquid biopsy refers to analyzing biomarkers in body fluids, playing an increasingly vital role in understanding cancer patients’ genomic landscapes, monitoring treatment responses, detecting minimal residual disease, and evaluating therapeutic resistance [95]. This technique offers a non-invasive alternative to traditional tissue biopsies, providing comparable molecular information without the need for invasive procedures [96]. Liquid biopsy allows for repeated monitoring of tumor biomarkers, helping assess tumor progression, guide personalized treatment, and evaluate therapeutic responses. It enables the collection of circulating tumor cells (CTCs), cell-free tumor DNA (ctDNA), RNA (ctRNA), proteins, peptides, and metabolites from a single sample, which can be used for various diagnostic and monitoring tests [97].

### Circulating Tumor Cells and cfDNA

CTCs are tumor cells released into peripheral blood, characterized by cytokeratin (CK)-positive and CD45-negative markers, and play a crucial role in tumor dissemination [98]. Studies have shown that MIBC patients treated with cisplatin-based NACT had lower CTC densities, with higher concentrations indicating better response rates [99]. For immunotherapy, monitoring PD-L1 expression on CTCs may optimize treatment with PD-L1 inhibitors, especially in non-responder BCG patients [100]. In non-metastatic bladder cancer patients post-cystectomy, HER2-positive CTCs benefited from targeted therapies, whereas HER2-negative CTCs indicated resistance against immunotherapy [101]. Additionally, the efficacy of immunotherapies, including checkpoint inhibitors, could be assessed through serial whole-blood CTC assays [102].

Most cell-free DNA (cfDNA) in blood is double-stranded, circulating as nucleosomes, which may exhibit tumor-specific histone modifications [103]. Bladder cancer patients’ cfDNA contains ctDNA, correlating with disease progression and poor outcomes [104]. Deep sequencing of ctDNA offers insights into the tumor genome, potentially predicting treatment response [105]. In a study of advanced UBC patients, cfDNA next-generation sequencing (NGS) identified genomic alterations comparable to tumor tissue, suggesting blood-based genomic screening as a non-invasive technique for identifying candidates for targeted therapies [106].

Sequencing of ctDNA in MIBC patients receiving NACT followed by radical cystectomy (RC), demonstrated that ctDNA presence before systemic therapy predicted worse RFS and OS [107]. Post-NACT, ctDNA-positive patients had a higher 12-month recurrence rate (75% vs. 11%). ctDNA analysis post-RC identified all patients who developed metastatic relapse, with 100% sensitivity and 98% specificity. Tumor subtype and immune signature analyses showed a significant mutational signature associated with ERCC2 status in NACT responders.

Studies of both plasma and urine cfDNA in MIBC patients before and during cisplatin-based NACT showed that patients with detectable ctDNA during systemic therapy experienced disease relapse [108]. Despite the potential, sequencing primary tumors and using ctDNA analysis for surveillance is time-consuming and expensive.

DNA methylation is increasingly recognized as a valuable biomarker for predicting response and prognosis in MIBC. Several studies have identified specific DNA methylation signatures that correlate with clinical outcomes in MIBC. A prognostic model based on 11 differentially methylated regions (DMRs) was developed, which demonstrated significant predictive value for survival in MIBC patients [109]. This model showed high predictive accuracy in both training and validation datasets, suggesting its potential utility in clinical practice for risk stratification and individualized therapy. Additionally, a three-gene methylation marker panel (KISS1R, SEPT9, and CSAD) that could predict nodal metastatic risk in MIBC has been identified [110]. This panel showed good sensitivity and specificity in differentiating between node-positive and node-negative tumors, which could guide decisions regarding extended lymph node resection and neoadjuvant chemotherapy. Xu et al. described a seven-probe DNA methylation signature that serves as an independent prognostic factor for overall survival in MIBC [111]. This classifier, referred to as a risk score (RS), was validated in a large cohort and significantly improved prognostic models when combined with clinical features. Recently, cfDNA methylation has been explored as a predictive biomarker for response to NACT in MIBC. Researchers developed a methylation-based response score (mR-score) that could predict the pathologic response to NACT, offering a minimally invasive method to guide treatment decisions [112].

## 7. Discussion

Urothelial bladder carcinoma remains a challenging malignancy due to its significant heterogeneity and varied treatment responses. This review highlights the importance of molecular biomarkers in guiding therapeutic decisions, particularly in advanced stages where traditional approaches offer limited benefit. Despite the advancements in understanding UBC’s molecular landscape, the integration of biomarkers into routine clinical practice has been slower compared to other malignancies like lung cancer.

The identification of molecular subtypes such as luminal, basal, and neuroendocrine-like UBC provides insights into potential therapeutic responses, with basal subtypes showing a better response to neoadjuvant chemotherapy. However, the clinical utility of these subtypes is still under investigation and requires further validation. Additionally, the role of DNA damage response and repair (DDR) genes, such as ERCC2 and ATM, in predicting responses to platinum-based therapies and immune checkpoint inhibitors offers promising avenues for personalizing treatment.

Emerging targeted therapies, including FGFR inhibitors and antibody–drug conjugates like enfortumab vedotin and sacituzumab govitecan, have expanded treatment options for patients with advanced UBC. However, the overall clinical outcomes remain modest, highlighting the need for continued research to refine biomarker-driven treatment strategies.

## 8. Conclusions and Future Directions

UBC exhibits high heterogeneity, but molecular studies offer longitudinal analysis opportunities for diagnosis, recurrence, progression, and treatment response prediction. With the abundance of genomic data from sequencing and expression studies, a current major obstacle is translating and applying this information into UBC clinical research. While many oncology clinics routinely incorporate tumor genomic testing, issues related to timing, cost, and interpretation remain significant hurdles. Ongoing trials aim to further integrate ICIs and ADC into routine bladder cancer treatment regimens. Understanding the interaction between ICIs and DNA-damaging agents is crucial, as DNA damage can both generate neoantigens and suppress immune responses. Integrative approaches combining genomic, proteomic, and functional data are needed to fully understand DNA repair dysfunction’s role in bladder cancer biology and treatment response.

## Figures and Tables

**Figure 1 cancers-16-03056-f001:**
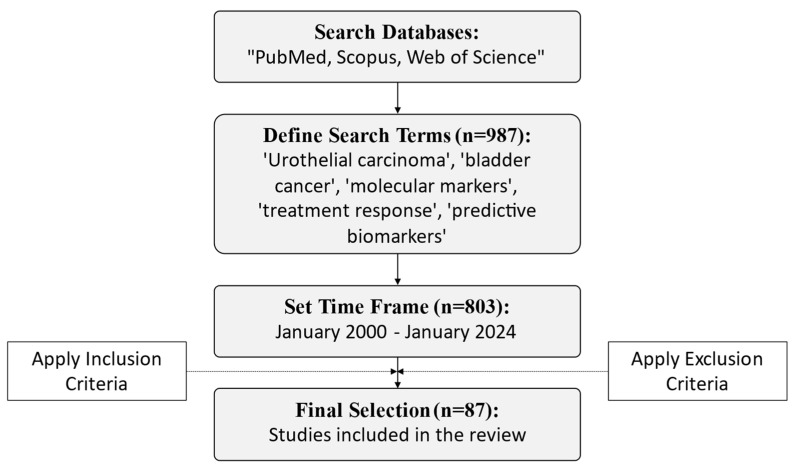
Flow diagram of the study selection process.

**Table 1 cancers-16-03056-t001:** Taxonomy of MIBC.

Differentiation	Urothelial/Luminal			Mixed	Basal	Neuroendocrine
Class name	Papillary	Non-specified	Unstable	Stromal-rich	Basal/squamous	Neuroendocrine-like
% of MIBC	24%	8%	15%	15%	35%	3%
Oncogenic mechanism	FGF3 + PPARG + CDKN2A +	PPARG +	PPARG + E2F3 + ERBB2 + Genomic instability Cell cycle +	-	EGFR +	TP53 – RB1 – Cell cycle +
Mutations	FGFR3 (~40%) KDM6A (~40%)	ELF3 (~35%)	TP53 (~75%) ERCC2 (~20%) TMB + APOBEC +	-	TP53 (~60%) RB1 (~25%)	TP53 (~95%) RB1 (~40%)
Stromal infiltrate	-	Fibroblasts	-	Smooth muscle Fibroblasts Myofibroblasts	Fibroblasts Myofibroblasts	-
Immune infiltrate	-	-	-	B-cells	CD8 T-cells NK cells	-
Histology	Papillary morphology (~60%)	Micropapillary morphology (~35%)	-	-	Squamous differentiation (~40%)	Neuroendocrine differentiation (~70%)
Clinical	T2 stage	Older patients (>80)	-	-	Women T3/4 stage	-
Median overall survival (yr)	4	1.8	2.9	3.8	1.2	1

**Table 2 cancers-16-03056-t002:** Promising DDR biomarkers associated with improved therapeutic response in MIBC.

Biomarker	Condition	Frequency	Pathway	Summary
ERCC2 [46]	Tissue testing from biopsied UBC before cisplatin-based NACT	~15%	Nucleotide excision repair	Alterations correlate with better response rates, pDS, and OS.
ATM [48]		~5%	Double-strand break repair	Mutations correlate with pCR and improved PFS, DSS, and OS.
RB1 [48]		~15%	Cell-cycle control	Alterations predict better response rates, PFS, DSS, and OS.
FANCC [48]		~10%	Homologous recombination repair	Mutations predict better response rates, PFS, DSS, and OS.
>1 gene [49]	Cisplatin-based NACT followed by cystectomy for MIBC	Varies	Double-strand break repair	Alterations in FANCD2, PALB2, BRCA1, or BRCA2 are associated with increased PFS in MIBC.

**Table 3 cancers-16-03056-t003:** Promising driver mutations associated with therapeutic response in UBC.

Biomarker	Condition	Frequency	Pathway	Summary
ERBB2	Tissue testing from biopsied UBC before cisplatin-based NACT	~5%	MAPK, PI3K	Mutations predicted pCR, pDS, and better CSS.
FGFR3	Tissue testing from biopsied UBC before cisplatin-based NACT	~50% overexpression; 15% alteration	MAPK, PI3K	Alterations correlated with pDS and worse PFS.
	Tissue testing from biopsied UBC before cisplatin-based adjuvant therapy			Mutations linked to worse PFS.
PIK3Ca	Tissue testing from biopsied UBC before cisplatin-based NACT	13–27%	PI3K	Alterations correlated with pDS.
HUS1	Tissue testing from biopsied UBC before cisplatin-based NACT	~1%	Mismatch repair	Amplification predicted non-response and worse PFS.
ABCA13	Tissue testing from biopsied UBC before cisplatin-based NACT	~5%	Mediation across cell membrane	Amplification predicted non-response and worse PFS.
EGFR	Tissue testing from biopsied UBC before cisplatin-based NACT	~70%	MAPK, PI3K	Alterations predicted non-response and worse PFS.
FIGNL1	Tissue testing from biopsied UBC before cisplatin-based NACT	~5%	Homologous recombination repair	Amplification predicted non-response and worse PFS.
IKZF1	Tissue testing from biopsied UBC before cisplatin-based NACT	~5%	Zinc finger transcription factor	Amplification predicted non-response and worse PFS.

**Table 4 cancers-16-03056-t004:** ICI trials in advanced urothelial cancer.

Trial	ICI	Setting	Response Rate in ITT	OS in ITT	PFS in ITT	PD1/PD-L1 Predictive Value
IMvigor210	Atezolizumab	1L, Cisplatin-ineligible	15%	15.9 months	2.7 months	Higher response rates (26%) in PD-L1-high tumors; IC2/3 ≥ 5%
Keynote-045	Pembrolizumab	2L, Post-platinum	21.1%	10.3 months	2 months	No significant difference in OS (8 months) or PFS (2.1 months) based on CPS ≥ 10%
IMvigor211	Atezolizumab	2L, Post-platinum	13%	8.6 months	2.1 months	No significant difference in OS (11.1 months) based on IC2/3 ≥5%
JAVELIN Bladder 100	Avelumab	1L, Maintenance post-chemo	16.1%	21.4 months	3.7 months	Improved OS (not reached) in PD-L1-positive tumors; expression in ≥25% of tumor cells
CheckMate 275	Nivolumab	2L, Post-platinum	19.6%	8.6 months	2.0 months	Higher response rates (28.4%) and superior OS (11.3 months) in PD-L1-positive tumors; PD-L1 expression ≥1%
Keynote-361	Pembrolizumab	1L	25.9%	15.6 months	Not reported	No significant difference in OS (16.1) based on CPS ≥ 10%
Keynote-361	Pembrolizumab	1L, Combination with chemo	25.9%	17 months	8.3 months	No significant difference in OS (17.0) based on CPS ≥ 10%
IMvigor130	Atezolizumab	1L	13%	15.2 months	2.7 months	Higher response rates (23%) and a trend toward improved OS (18.6 months) in IC2/3 ≥ 5%
CheckMate 032	Nivolumab	2L, Post-platinum	25.6%	9.7 months	2.7 months	No significant difference in response rates (26.9%) and OS (11.3 months) in PD-L1-positive tumors; PD-L1 expression ≥1%
CheckMate 032	Nivolumab + Ipilimumab	2L, Post-platinum	26.9%	7.3 months	2.6 months	Higher response rates (38%) and improved OS (15.3 months) in PD-L1-positive tumors; PD-L1 expression ≥1%
PURE01	Pembrolizumab	Neoadjuvant	42% (pCR)	36-month OS was 83.8%	36-month EFS was 74.4%	Higher pCR rates (54.3%) in PD-L1-positive tumors; CPS ≥ 10%
DANUBE	Durvalumab + Tremelimumab	1L, Combination with chemo	24%	12.9 months	5.5 months	No significant OS benefit in (14.4 months) TC ≥ 25% compared to ITT population
EV-302	Pembrolizumab + Enfortumab Vedotin	1L	67.7% (29.1% pCR)	31.5 months	12.5 months	Higher response rates (73.3%) and improved OS (not reached) in PD-L1-positive tumors; CPS ≥ 10%

TPS, Tumor positive score; CPS, Combined positive score; TC, Positivity in tumor cells; IC, Positivity in tumor-infiltrating immune cells.

**Table 5 cancers-16-03056-t005:** Promising miRNAs associated with therapeutic response in UBC.

Category	miRNA	Target/Regulator	Function
Promoting Chemosensitivity	miR-7-5p [79]	ATG7	Inhibits invasive characteristics and enhances chemosensitivity.
	miR-30a-3p [80]	MMP2 and MMP9	Improves apoptosis and reduces cell viability when combined with cisplatin and decreases migration and invasion.
	miR-31 [81]	ITGA5	Enhances chemosensitivity to mitomycin-C and inhibits proliferation, migration, and invasion.
	miR-34a [82]	TCF1, LEF1, Cdk6, SRT-1, CD44	Enhances sensitivity to epirubicin and cisplatin while repressing metastatic characteristics.
	MiR-101[83]	COX2	Promotes chemosensitivity to cisplatin.
	MiR-101-3p [84]	EZH2, affects MRP1 expression	Enhances sensitivity to gemcitabine.
	miR-129-5p [85]	Wnt5a	Promotes response to gemcitabine.
	miR-27a [86]	SLC7A11	Increases cisplatin sensitivity.
	miR-642 [86]	Unknown	Increases cisplatin sensitivity.
	miR-34a [82]	Unknown	Increases cisplatin and epirubicin sensitivity.
	Cdr1as [87]	Unknown	Increases cisplatin sensitivity through the miR-1270/APAF1 axis.
Promoting Chemoresistance	miR-21 [88]	PTEN	Promotes resistance to doxorubicin and inhibits doxorubicin-induced apoptosis.
	miR-22-3p [89]	NET1	Enhances chemoresistance by increasing cell viability and colony formation while reducing apoptosis.
	miR-93 [90]	Unknown	Promotes chemoresistance without direct binding to LASS2.
	miR-98 [91]	LASS2	Leads to increased proliferation, resistance to cisplatin and doxorubicin, and reduced apoptosis.
	miR-130b [92]	CYLD	Promotes chemoresistance.
	miR-193a-3p [93]	Unknown	Promotes multi-chemoresistance.
Correlated with Better Response and Survival	miR-886-3p [86]	Unknown	Associated with complete response (CR) and better overall survival (OS) in metastatic cases treated with MVAC or Gem-Cis.
	miR-923 [86]	Unknown	
	miR-944 [86]	Unknown	
	miR-203 [94]	Unknown	Low expression is correlated with worse progression-free survival (PFS) and OS.
Correlated with Worse Response and Survival	miR-372 [88]	Unknown	High expression is linked to worse PFS.
	miR-21 [88]	Unknown	High expression is associated with shorter PFS in metastatic cases treated with MVAC or Gem-Cis.
Mixed Responses Based on Expression Levels	miR-138 [86]	Unknown	Decreasing expression increases cisplatin sensitivity.
	miR-101 [83]	Unknown	Downregulation induces cisplatin resistance through the COX-2 axis.

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
