# Peer review of "Contemporary Molecular Markers for Predicting Systemic Treatment Response in Urothelial Bladder Cancer: A Narrative Review"

_cancers, 2024, doi:10.3390/cancers16173056_

Round 1
Reviewer 1 Report
Comments and Suggestions for Authors
This is a well written, contemporary summarized of current therapeutic biomarkers associated with response to bladder cancer treatment. The authors provided a nice overview of the landscape of treatment for bladder cancer and the dived into the biomarkers associated with each treatment, including both tissue based and circulating ones. The entire review is organized in a logical manner that is easy to follow. No major comments.
Minor comment:
- Please provide a simple summary of the manuscript as required by the journal.
Author Response
- Please provide a simple summary of the manuscript as required by the journal.
Thank you for the positive feedback. A short simple summary has been provided as per journal requirements.
Reviewer 2 Report
Comments and Suggestions for Authors
- Please provide a Simple Summary
- The introduction effectively explains the context and motivations for this review on predictive molecular markers of therapy response in bladder urothelial carcinoma, in light of the current research landscape in this field.
- The methodology is clearly and comprehensively explained; the authors are encouraged to add a brief paragraph in this section regarding the approach used to analyze and/or synthesize the articles.
- Section 4. The section is complete and well-structured; the authors are encouraged to further elaborate on the discussion of the limitations of these therapies, particularly concerning patient selection.
- Section 5-subsection 5.1. The authors are encouraged to mention and discuss the possibility of using surrogate markers for molecular classification of bladder carcinoma on paraffin-embedded material (10.3390/ijms23147819; 10.1111/pin.13060; 10.3390/cancers12071784). Additionally, the authors are invited to add a paragraph discussing the application of these markers in clinical practice.
- Sections 7-8. The authors are encouraged to discuss the importance of using predictive markers in NMIBC, on the basis of current research (doi: 10.3390/jcm13082182; 10.1038/s41585-024-00914-7; 10.1111/iju.15370).
Author Response
- Please provide a simple summary of the manuscript as required by the journal.
Thank you for the positive feedback. A short simple summary has been provided as per journal requirements.
- The methodology is clearly and comprehensively explained; the authors are encouraged to add a brief paragraph in this section regarding the approach used to analyze and/or synthesize the articles.
Added a short paragraph with more details regarding the analysis of selected papers and synthesis of the final review as per request.
- Section 4. The section is complete and well-structured; the authors are encouraged to further elaborate on the discussion of the limitations of these therapies, particularly concerning patient selection.
Added a short sub-section (4.5) regarding the limitations of current systemic therapies and patient selection as per request.
- Section 5-subsection 5.1. The authors are encouraged to mention and discuss the possibility of using surrogate markers for molecular classification of bladder carcinoma on paraffin-embedded material (10.3390/ijms23147819; 10.1111/pin.13060; 10.3390/cancers12071784). Additionally, the authors are invited to add a paragraph discussing the application of these markers in clinical practice.
Added a short paragraph highlighting the importance of molecular subtyping via using surrogate markers while citing recommended sources.
- Sections 7-8. The authors are encouraged to discuss the importance of using predictive markers in NMIBC, on the basis of current research (doi: 10.3390/jcm13082182; 10.1038/s41585-024-00914-7; 10.1111/iju.15370).
We appreciate your observation regarding the focus on muscle-invasive bladder carcinoma (MIBC) and the need to address non-muscle invasive carcinoma (NMIBC). However, it is important to note that NMIBC is typically managed with local treatments, such as surgical resection followed by intravesical instillations of Bacillus Calmette-Guérin (BCG) or chemotherapy, rather than systemic therapies. As our review focuses on predictors of response to systemic treatments, we did not include NMIBC in this narrative review.